# Workplace Loneliness and the Need to Belong in the Era of COVID-19

**Shuyun Du [1], Yinan Ma [1,2,\*] and Jeoung Yul Lee [2,3]**

[1] School of Tourism Management, Zhengzhou University, Zhengzhou 450001, China; dsy@zzu.edu.cn
[2] School of Business Management, Hongik University, Sejong 30016, Korea; jeoungyul@hongik.ac.kr
[3] National Research Base of Intelligent Manufacturing Service, Chongqing Technology and Business University, Chongqing 400067, China
[\*] Correspondence: zchmyn@gs.zzu.edu.cn

**Abstract:** On the basis of Social Exchange Theory (SET), Knowledge Stickiness Theory (KST), and the need-to-belong hypotheses, we empirically studied the causes and consequences of workplace loneliness in interpersonal communication and explored the moderating effect of the need to belong. We distributed a survey and collected 639 valid responses in mainland China in both paper and electronic form during the period of February to October 2020, when the COVID-19 crisis was severe. Mplus was used to create a latent structural equation model with a moderating mediating model. Collaborative and competitive intrateam climates affect employees' workplace loneliness and knowledge hoarding from different aspects. We also verified a moderated mediation model. Thus, this study examines the mediating effect of workplace loneliness and introduces the need to belong as the moderating variable; reveals the formation mechanism of workplace loneliness in collaborative and competitive intrateam climates; and deepens the research on the effective regulation of workplace loneliness. As the COVID-19 pandemic remains ongoing, we have verified changes to the mediating effect of workplace loneliness, driven by the motivation of the need to belong, and clearly evaluated a moderated mediating effect path, which contributes to the theory of belonging.

**Keywords:** collaborative intrateam climate; competitive intrateam climate; workplace loneliness; employee relationship; interpersonal relationship; need to belong; knowledge hoarding; COVID-19; China

## 1. Introduction

As loneliness is a fundamental part of the human experience, everyone may have experienced it to some extent [1]. According to a 2016 Harris poll, nearly three-quarters of Americans experience loneliness every year and on an ongoing basis, with as many as one-third saying they feel lonely at least once a week [2,3]. Loneliness has been described as an unpleasant feeling [4]. Although loneliness has been widely studied from the perspective of clinical psychology, the concept of workplace loneliness remains underexplored and under-investigated [4], especially during COVID-19. The pandemic has also raised awareness that communication and efficient responses are essential [5].

The concept of workplace loneliness has attracted attention since 2006 and, in particular, since 2016. Studies from a number of countries have shown that workplace loneliness is not limited to one geographical location but a phenomenon that affects people and economies around the world [3]. Research on loneliness in the workplace is meaningful because individuals spend as much time at work as they do at any other location [6]. Moreover, the loneliness generated directly or indirectly in the workplace will negatively impact employees' social and professional life [7,8], particularly in the era of COVID-19.

In recent years, especially during the COVID-19 crisis, loneliness in the workplace has become an increasingly serious issue and has attracted growing attention due to its complex consequences [4]. The experience of loneliness in the workplace will quickly

weaken employees' close relationship with the group, which is likely to produce the "island effect" [7]. In addition, researchers have proved that workplace loneliness negatively affects employees' attitudes and reduces employee commitment [8,9]. In other words, if an employee experiences loneliness in the workplace, they are subject to low energy levels, which may reduce their commitment to the organization [8,10]. Some studies have theoretically proposed that workplace loneliness is contagious; that is, individual loneliness in the organization will affect other members through the emotional contagion effect and reduce individual relationship performance and team performance [11,12]. In terms of behavioral performance, loneliness leads to the decline of in-role performance and out-of-role behavior [13] and even induces turnover intention among employees [9], which is even more critical in the era of COVID-19.

Some studies have verified that knowledge hiding has a negative impact on creativity and plays a mediating role between workplace loneliness and creativity [4]. Another study found that low-quality team member communication may further aggravate employees' sense of loneliness and alienation in the team, resulting in more silent behaviors at work [1]. Knowledge hoarding and knowledge hiding, which belong to workplace silence, are often compared. Despite the many encouraging findings about knowledge sharing in previous studies, knowledge hoarding still exists among organization members [14–16]. Most of the time, these organization members work against the mutually beneficial interests of the workplace and undermine peer camaraderie and support. Hoarding behavior, which impedes the flow of knowledge transfer, has been found to impair performance at the interpersonal and organizational levels [17,18]. In contrast to the effects on individuals and organizations of knowledge hoarding, its antecedents have not been widely studied [16,19]. This paper will study the antecedents of knowledge hoarding, especially the influence of workplace loneliness on knowledge hoarding in the era of COVID-19.

The concept of knowledge management is regarded as an important managerial activity [20–22]. However, with the continuous expansion of the research scale of knowledge sharing, anti-productive knowledge behavior has received little attention [22]. Others have commented that a better understanding of the antecedents and processes of non-sharing behavior may be the key to explaining why many companies are not successful in implementing knowledge-sharing efforts [18,19,23]. It is speculated that the lack of knowledge sharing within an organization is probably due to problems in other aspects, such as the working environment [23], especially in times of uncertainty such as COVID-19. Environmental factors are working conditions that hinder the establishment of effective working relationships with colleagues, thereby inhibiting knowledge transfer [18,24].

Like knowledge hoarding, organizational environments such as those under COVID-19 have been listed by many scholars as among the antecedents of workplace loneliness. Workplace loneliness is defined as employees' subjective emotional evaluation and feeling of whether their partners and work institutions meet their needs for sense of belonging [25,26]. This workplace-specific emotion is closely related to the interpersonal environment in the workplace [25,26].

One plausible assumption is that in cultures that emphasize cooperation and relatedness, the quality of relationships may be healthier and there may be less loneliness [27,28], confirming that a competitive and uncooperative organizational environment is the cause of workplace loneliness. When the climate of an organization is strongly competitive, employees pay more attention and are more sensitive to, leadership relationship resources. In such a climate, employees with lower comparative results are more likely to have a sense of relative deprivation, resulting in a stronger sense of workplace loneliness [29]. In sum, we believe that it is reasonable and meaningful to explore the antecedents of workplace loneliness and knowledge hoarding in the organizational environment, especially in terms of team cooperativeness and competition.

Yet, a survey on the relationship between videoconferences and fatigue due to the COVID-19 found that a higher sense of belonging was associated with less silence, and a lower sense of belonging was associated with more silence [30]. As far as we know, no

comprehensive study has been carried out to address this question. This paper tries to adjust and improve the negative impact of organizational climate on workplace loneliness and knowledge hoarding by the need to belong to generate a closer relationship between employees and better organizational performance. The rest of the paper is structured as follows. Section 2 will explore the effect of organizational climate on workplace loneliness and knowledge hoarding and discuss the mediating role of workplace loneliness and the moderating effect of the need to belong. In this process, we reasonably examined a moderated mediation effect. Sections 3 and 4 present the research methods and analysis of the research results. The paper concludes with a brief discussion and summary.

## 2. Theoretical Background and Hypotheses

Team members are often motivated to compete and collaborate simultaneously [31,32]. Similarly, George found in a study on the necessity of studying the influence of individuals and groups on creativity that, on the one hand, the motivation to meet relevant needs may lead to a collaborative team climate. On the other hand, individual performance rewards encourage team members to compete and create a competitive team atmosphere [33,34]. Unfortunately, the impact (good or bad) of such intrateam cooperation and competition on the creativity of individual team members is still an empirical problem [32]. This paper predicts that a collaborative and competitive intrateam climate will affect workplace loneliness and knowledge hoarding to varying degrees, respectively, in the era of COVID-19.

### 2.1. Collaborative Intrateam Climate, Competitive Intrateam Climate, Workplace Loneliness, and Knowledge Hoarding

We use Social Exchange Theory (SET) to understand team climate and workplace loneliness. Organizational climate is employees' perception of the quality of the internal environment of the organization, which plays an important role in the formation and development of interpersonal relationships at work [35]. An empirical survey showed that community spirit is significantly negatively correlated with workplace loneliness [12]. We define collaboration within a team as the intentional sharing and acceptance of individual efforts, knowledge, and resources with other team members to achieve a common goal [32,36]. The core of SET is based on the principle of reciprocity. When one party provides help or resources to the other, the helpee has an obligation to repay the helper so as to establish the ethics of social exchange [1,37]. It is the basic psychological need of all people to experience relationships by caring for each other and feeling the non-contingent value of others [26,38].

Employees are likely to feel lonely in the workplace if they cannot handle the complicated interpersonal relationships there [1]. However, in contrast, employees in a collaborative climate help other colleagues in the team or provide other supportive and positive team resources (good emotional communication, interpersonal and team support, etc.) [1]. The communication environment of mutual help and friendly cooperation will make employees feel better while working and reduce their loneliness in the workplace. Therefore, it is reasonable to propose the following hypothesis:

**H1a.** *A collaborative intrateam climate is negatively associated with workplace loneliness.*

Wright pointed out that there are all kinds of complicated interpersonal relationships in the workplace, and employees are likely to feel lonely if they cannot deal with these relationships well [35]. Fierce competition in the workplace makes it difficult to maintain sincere communication among organization members. It is difficult to form friendships in the workplace when the organizational climate emphasizes rewards, punishment, and personal interests while ignoring teamwork and mutual trust [35]. The problem of workplace loneliness has become increasingly prominent. Individuals with a strong sense of competition desire to win over and surpass others, and it is difficult to trust and form intimate relationships with others. Therefore, a competitive mentality is significantly positively correlated with loneliness in the workplace [12].

Previous studies have found that when there is a strongly competitive atmosphere in an organization, employees attach more importance to, and are more sensitive to, leadership relationship resources. In such circumstances, employees with lower comparative results are more likely to feel relative deprivation and thus have stronger workplace loneliness [29]. In a highly competitive department or team, the rewards obtained by individuals depend on their performance compared to other members. In addition, leaders often compare employees' performance with their colleagues, so employees experience a strong sense of competition and pressure.

The perception of a competitive climate will affect individuals' cognition of and attitude towards resources [39,40]. In a different competitive climate, employees have different desires for leadership relationship resources and pay different attention to leadership member exchange and social comparison. Kilduff et al. pointed out that competition among employees can arouse individuals' concern about their status and self-worth within an organization [29,41]. From the perspective of social comparison, we believe that there is a strong relationship between a competitive climate within the team and workplace loneliness:

**H1b.** *A competitive intrateam climate is positively associated with workplace loneliness.*

In the strategic management literature, stickiness is the basis of withholding and hoarding behaviors. To promote knowledge-management literature, one study has explained the factors affecting salespeople's knowledge-hoarding behavior under the guidance of Knowledge Stickiness Theory (KST) [18]. By definition, stickiness represents the difficulties associated with knowledge transfer. Szulanski described stickiness as the behavior of slowing down and hindering the flow of knowledge within enterprises [18,24,42]. Stickiness indicates an individual's control over the information they have [43]. When employees are worried about losing ownership of proprietary information, stickiness reinforces an individual's tendency to withhold, where the transfer of knowledge is hampered by the intent of the source and the environmental characteristics surrounding the source or situation [44]. Szulanski also suggested that stickiness is a difficult process of knowledge transfer, from initial persistence to withholding knowledge and then to more conventional behavior, which is carried out without conscious thought processing or a lower degree of intention (hoarding) [42].

According to Szulanski's hypothesis, lack of motivation for knowledge transfer may lead to knowledge viscosity, but empirical results show that this influencing factor is not obvious [24]. Feng and Liao analyzed that on the one hand, the incentive system for employees of the eight famous companies selected by Szulanski is appropriate, and the result is not representative of most ordinary companies [24,45]. On the other hand, when filling in a questionnaire survey, respondents may have the tendency to avoid mention of their own shortcomings; that is, they may attribute the causes of knowledge viscosity to objective factors [45]. Feng and Liao summarized the three causes of knowledge stickiness as cognitive factors, knowledge transfer environment, and transfer motivation [45]. Knowledge is essentially human logic, and human motivation is the main source of stickiness. In other words, the motivation for the knowledge transfer of organizational members affects the viscosity of knowledge. The stronger the motivation for knowledge transfer, the lower the viscosity of knowledge [45].

In a fierce and competitive environment, communication becomes more chaotic [5]. In a purely competitive climate, team members will try to surpass other members to obtain external rewards. Members have the motivation to perform better than others in the team and are eager to show a high level of performance. A highly competitive team climate creates highly external motivation [32]. According to resource conservation theory, it is in the personal interest of employees to hoard knowledge. In a highly competitive work environment, knowledge is often regarded as a valuable competitive commodity that should not be shared casually [46,47]. When driven by this motivation, the knowledge stickiness of employees will be enhanced.

Similarly, a collaborative intrateam climate describes shared perceptions among team members. Cooperation is considered to have a positive impact on intrinsic motivation [32,48]. Positive interaction between team members can lead to the satisfaction of relevant needs and then produce internal motivation. In addition, the ideas, information, and viewpoints exchanged in the process of cooperation enrich one's knowledge base [49], which is contrary to the direction of knowledge hoarding. Taken together, we predict the relationship between intrateam climate and knowledge hoarding thus:

**H2a.** *A collaborative intrateam climate is negatively associated with knowledge hoarding.*

**H2b.** *A competitive intrateam climate is positively associated with knowledge hoarding.*

Workplace loneliness hinders effective communication among organization members and causes a series of negative impacts on them, the team, and even the organization. Lam and Lau pointed out that workplace loneliness is negatively correlated with organizational citizenship behavior [13]. Based on SET, individuals hope to establish a stable trust relationship with others to achieve long-term reciprocity. Employees will not only do their own work effectively but also make extra efforts to achieve organizational goals when the organization meets their needs for intimacy and social communication. Lonely employees often deal with risks in a negative way, reducing altruistic behavior in the organization due to their lack of trust in others. Meanwhile, poor self-evaluation will lead them to be reluctant to seek new social relations. They pay more attention to negative than positive social information, which will lead to their reluctance to engage in social exchange in the workplace [7,50,51].

Workplace loneliness negatively affects team member exchanges [1]. Low-quality team member exchanges can make employees avoid socializing. They will remain silent about problems and refrain from making work suggestions in the team due to a lack of self-confidence and support from team members [1]. Interpersonal relationships, social exchange, and distrust may influence knowledge-hiding behavior [52]. Since knowledge hoarding and knowledge hiding have very similar definitions, this behavior runs counter to organizational citizenship behavior and belongs to the negative and silent response. In sum, this paper proposes the following hypothesis:

**H3.** *Workplace loneliness is positively associated with knowledge hoarding.*

Workplace loneliness is an individual's experience of being deprived of their need to control interpersonal relationships and their sense of existence in an organization. In essence, it is an individual's feelings surrounding workplace interpersonal relationships and organizational self-esteem [53]. Wright put forward the following theory on workplace loneliness: The work response of more lonely employees will be more affected by their organizational environment [35]. Studies have shown that lonely employees tend not to actively seek out social relationships but use passive avoidance, an arrogant coping strategy [50,54]. The main reason for workplace loneliness and knowledge hiding is the lack of interpersonal relationships in the workplace. Based on the above analysis, we assume:

**H4a.** *Workplace loneliness mediates the relationship between a collaborative intrateam climate and knowledge hoarding.*

**H4b.** *Workplace loneliness mediates the relationship between a competitive intrateam climate and knowledge hoarding.*

### 2.2. The Moderating Role of Need to Belong

People have the basic need to connect with others [55]. In their landmark article, Baumeister and Leary proposed that the need to belong is a powerful, universal, and influential human driving force [55]. The authors collected a large amount of evidence showing that "all humans have a universal driving force to form and maintain at least a minimum number of lasting, positive and influential interpersonal relationships". Weiss predicted that people need both regular and continuous relationships. People believe that

the feeling of loneliness can be exacerbated by insufficient social contact (social loneliness) or lack of meaningful intimacy (emotional loneliness) [55]. Loneliness arises when the need to belong is not met [55,56]. Baumeister and Leary argue that to fulfil the need to belong, both quantity and quality of interaction must be satisfied [55].

An important implication of the need to belong is that it "transforms people from a self-directed model to a more cooperative and collective response model" ([55], p. 519). Another study similarly showed that when facing the dilemma of public goods, people with a high need to belong are more likely to cooperate and show helpful behavior to others [57,58]. Those with a sense of belonging are not only more helpful but also more self-sacrificing to the group [59]. In addition, people with a long-term high sense of belonging are more sensitive to social cues [59]. They can recognize voice intonation and facial emotions more accurately and show better social skills [60,61]. Finally, experimental evidence shows that individuals with a high sense of belonging tend to think that others are more similar to themselves [62], while people with a high need to belong are motivated to build a community that meets their need to belong [61].

The pursuit of a sense of belonging urges individuals to eliminate the negative states associated with exclusion and pain [63,64]. Studies have found that the need to belong can reduce the sense of alienation, isolation, and dissatisfaction or low sense of social integration and social exclusion [65–67]. Other research results show that the desire for acceptance and belonging can moderate interpersonal behavior in important aspects [68]. For example, people with a high need to belong show a stronger willingness and ability to cooperate in teamwork, probably because cooperation helps establish a good member image [57,68]. Martončik and Lokša found that guild members in MMORPGs feel less loneliness than non-guild members because the guild provides players with a sense of belonging and opportunities to cooperate [69,70]. Kramer and Brewer showed that when the sense of belonging is stimulated by highlighting group identity, people are more likely to restrain their egoistic tendencies and cooperate with others for the greater interests of the group [55,71].

The idea of this paper may be consistent with previous research suggesting that the need to belong moderates the perceived team climate and workplace loneliness. The need for belonging has many benefits for mental health and can enable people to obtain acceptance and avoid rejection [61], improve social inclusion and life satisfaction, and reduce social exclusion [67]. When working in a collaborative intrateam climate, employees with a high sense of belonging are more motivated to maintain close and friendly communication with colleagues and are less likely to feel lonely in the workplace. At the same time, the need for a sense of belonging seems to overcome some tendencies of hostility, competition or division [55] and reduce the sense of alienation and isolation [67]. When the competitive intrateam climate is strong and considering the need to belong, employees with a high sense of belonging will adjust their negative emotions and behaviors to reduce hostility with colleagues, increase cooperation and support with colleagues, and alleviate the loneliness in the workplace. To this end, we propose the following hypothesis:

**H5a.** *The need to belong weakens the negative relationship between collaborative intrateam climate and workplace loneliness.*

**H5b.** *The need to belong weakens the positive relationship between competitive intrateam climate and workplace loneliness.*

We regard knowledge as a kind of information. Knowledge sharing, knowledge hiding or knowledge hoarding are different ways for people to deal with information. Social connection creates a model in cognitive processing which gives priority to organizing information for people with some connection with themselves. Several studies have shown that people deal with intimate partner information differently from information about strangers [55]. Clark showed that people have different ways to track information about potential partners [72]. In addition, research by Tice et al. showed that people tend to be more modest when presenting information to friends (compared with strangers) [73].

Individuals with a high sense of needing to belong are more motivated to obtain social connections than those with a low level [74]. Based on the above, we infer that when employees with a high sense of needing to belong experience less workplace loneliness because they obtain more satisfactory social connections, and they process information with a more positive and open attitude, so as to reduce knowledge hoarding. On this basis, we speculate a moderated mediating effect:

**H6a.** *The need to belong positively moderates the mediating effect of workplace loneliness on the relationship between collaborative intrateam climate and knowledge hoarding.*

**H6b.** *The need to belong negatively moderates the mediating effect of workplace loneliness on the relationship between competitive intrateam climate and knowledge hoarding.*

Our main assumptions are tested and reported below (Figure 1). A moderated mediation analysis enables a better understanding of the conditions under which need to belong moderates the impact of team climate on workplace loneliness and knowledge hoarding.

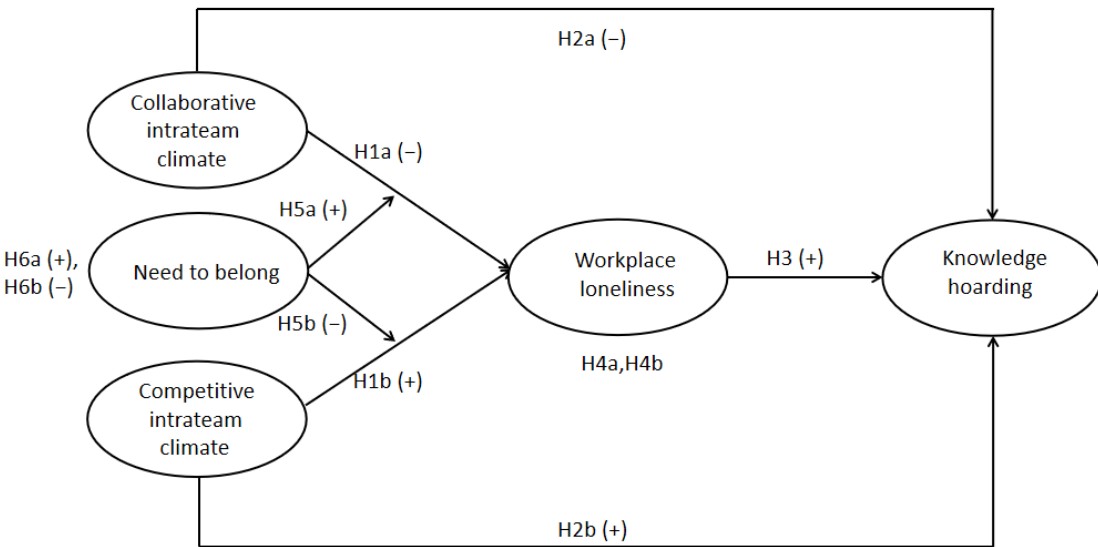

**Figure 1.** Conceptual model.

### 3. Method

*3.1. Research Design and Sample*

We adopted the research method of administering a questionnaire survey, which was distributed and collected in mainland China during the COVID-19 crisis in both paper and electronic form. The questionnaires were distributed three times, from 4 February to 11 February 2020, from 12 February to 22 February 2020, and from 1 October to 10 October 2020. Respondents were both self-employed and employed, working in industries including hospitals, schools, state-owned enterprises, and others, and it was stipulated that there must be more than five participants in the department. A total of 639 valid samples were obtained for our study.

In order to effectively ensure the quantity and quality of the data, we adopted three methods: (1) By seeking the help of friends from different industries around us and training them on the operation process, we successfully recruited several assistants for questionnaire distribution, thus improving its efficiency and expanding its scope. (2) By leveraging the collective activities carried out in the respondents' units, we distributed and collected the questionnaires. (3) We effectively enhanced the enthusiasm of the respondents by giving them gifts. These methods helped us achieve the expected goal, and the effective recovery rate of the questionnaire was 94%.

The basic characteristics of the respondents are shown in the table (Table 1) below.

**Table 1.** Basic features of respondents (*n* = 639).

| Variable | Attribute | Frequency | Percent |
|---|---|---|---|
| Age (Years) | 20–30 | 419 | 65.6 |
| | 31–40 | 142 | 22.2 |
| | 41–50 | 45 | 7 |
| | 51–60 | 33 | 5.2 |
| Gender | Male | 244 | 38.2 |
| | Female | 395 | 61.8 |
| Education | High school and below | 39 | 6.1 |
| | Junior college | 78 | 12.2 |
| | Undergraduate | 361 | 56.5 |
| | Master's degree or above | 161 | 25.2 |
| Post | Managers | 135 | 21.1 |
| | Scientific and technical personnel | 103 | 16.1 |
| | Marketers | 94 | 14.7 |
| | Administrative personnel | 218 | 34.1 |
| | Production personnel | 89 | 13.9 |
| Post | Top management | 17 | 2.7 |
| | Middle and general management | 182 | 28.5 |
| | Employee or other | 440 | 68.9 |

*3.2. Statistical Methodology*

SPSS 26.0 and Mplus 8.3 were used for statistical analysis. Mean and standard deviation are used to describe continuous variables. Frequency and percent are used to describe categorical variables. The Harman single factor test was used to test common method deviation. Cronbach's Alpha (Cronbach coefficient $\alpha$) was used as the reliability measurement index. Exploratory factor analysis and confirmatory factor analysis were used to explore the validity of the scale. Composite reliability and average variance extracted (AVE) were used to detect combination reliability and convergence validity, respectively. Correlation analysis was conducted to investigate the relationship between variables. The AVE square value being > the correlation coefficient between this variable and other variables indicated that this variable had good discriminant validity. Mplus was used to establish a latent structural equation model to explore the mediating effect of workplace loneliness on collaborative intrateam climate, competitive intrateam climate, and knowledge hoarding. The need to belong moderated the mediating effect of workplace loneliness between collaborative intrateam climate, competitive intrateam climate, and knowledge hoarding.

*3.3. Common Method Bias Test*

The Harman single factor test adopted in this study is frequently used to test for common method bias. If the variance explanation percentage of the unrotated first common factor in principal component factor analysis is less than 50% [75], it is considered that there is no serious common method bias. A Harman single factor test was conducted on all the scale items of the questionnaire, and a total of six common factors with characteristic values greater than 1 were extracted. The cumulative variance explanation degree was 64.747%, indicating that the principal component analysis method could better cover the main information, and the variance explanation rate of the first factor without rotation was 30.607%, less than 50%. Therefore, there is no serious common method deviation in the questionnaire.

*3.4. Reliability Analysis and Validity Analysis*

In this study, Mplus was used to establish a first-order six-factor confirmatory factor analysis model, and the model fitting index was: $\chi^2$ = 1252.98, DF = 579, $\chi^2$/DF = 2.164 < 5, RMSEA = 0.043 < 0.08, SRMR = 0.045 < 0.08, CFI = 0.947 > 0.9, TLI = 0.942 > 0.9. All indexes reached the fitting standard, indicating that the confirmatory factor analysis model

can be supported by data and is well structured. The reliability analysis results of each variable/scale, combined reliability, and average method deviation extraction results are showed in the Table 2:

**Table 2.** Reliability analysis and validity analysis (*n* = 639).

| Variable | Cronbach's $\alpha$ | CR | AVE |
|---|---|---|---|
| CLIC | 0.899 | 0.9 | 0.692 |
| CPIC | 0.883 | 0.885 | 0.525 |
| ED | 0.929 | 0.929 | 0.594 |
| SC | 0.893 | 0.898 | 0.558 |
| NTB | 0.841 | 0.844 | 0.52 |
| KH | 0.817 | 0.819 | 0.532 |

Collaborative intrateam climate/CLIC contains four items: Cronbach's $\alpha$ is 0.899, the combined reliability value is 0.900 > 0.7, and the mean variance extraction value is 0.692 > 0.5. Competitive intrateam climate/CPIC contains seven items in one dimension, Cronbach's $\alpha$ is 0.883, the combined reliability value is 0.885 > 0.7, and the mean variance extraction value is 0.525 > 0.5. Workplace loneliness/WL includes two dimensions: emotional deprivation/ED and social companionship/SC, with a total of 16 questions. Cronbach's $\alpha$ was 0.929 and 0.893, the combined reliability values were 0.929 and 0.898 > 0.7, and the mean variance extraction values were 0.594 and 0.558 > 0.5, respectively. The need to belong/NTB contains five items in one dimension, Cronbach's $\alpha$ is 0.841, the combined reliability value is 0.844 > 0.7, and the mean variance extraction value is 0.520 > 0.5. There were four items of knowledge hoarding/KH with one dimension, Cronbach's $\alpha$ was 0.817, combination reliability value was 0.819 > 0.7, and the AVE value was 0.532 > 0.5.

### 3.5. Discriminant Validity

In this study, confirmatory factor analysis was used to test the discriminant validity among variables. A confirmatory factor model was established by Mplus, and a six-factor model was used as the benchmark model. As shown in the Table 3, the model fitting indexes were $\chi^2$/DF = 2.164 < 5, RMSEA = 0.043 < 0.08. As CFI = 0.947 > 0.9 and TLI = 0.942 > 0.9, all of them reached the standard [76]. The fitting degree is better than in other models, indicating that the variables have good discriminant validity.

**Table 3.** Discriminant validity (*n* = 639).

| Fit | Model | $\chi^2$ | Df | $\chi^2$/df | RMSEA | SRMR | CFI | TLI |
|---|---|---|---|---|---|---|---|---|
| 6 Factor | CLIC, CPIC, ED, SC, NTB, KH | 1252.980 | 579 | 2.164 | 0.043 | 0.045 | 0.947 | 0.942 |
| 5 Factor | CLIC, CPIC, ED + SC, NTB, KH | 2721.811 | 584 | 4.661 | 0.076 | 0.072 | 0.832 | 0.818 |
| 4 Factor | CLIC + CPIC, ED + SC, NTB, KH | 4148.582 | 588 | 7.055 | 0.097 | 0.096 | 0.720 | 0.700 |
| 3 Factor | CLIC + CPIC + ED + SC, NTB, KH | 5247.923 | 591 | 8.880 | 0.111 | 0.101 | 0.633 | 0.609 |
| 2 Factor | CLIC + CPIC + ED + SC + NTB, KH | 6354.278 | 593 | 10.715 | 0.123 | 0.117 | 0.546 | 0.518 |
| 1 Factor | CLIC + CPIC + ED + SC + NTB + KH | 6853.971 | 594 | 11.539 | 0.128 | 0.120 | 0.507 | 0.477 |
| Criteria | | | | <5 | <0.08 | <0.08 | >0.9 | >0.9 |

### 3.6. Correlation Analysis

Correlation analysis tests the correlation according to the correlation coefficient between variables. The Pearson coefficient is a commonly used correlation coefficient in social science research. If the correlation coefficient passes the significance test, it indicates that there is a statistically positive or negative correlation between variables. In contrast, if the correlation coefficient fails to pass the significance test, it indicates there is no statistical correlation between variables. As shown in the Table 4, values are as follows: correlation coefficient: <0.4, weak correlation; 0.4–0.7, medium-intensity correlation; >0.7, high correlation; >0, positive correlation; and <0, negative correlation.

**Table 4.** Interscale correlation (*n* = 639).

| | Mean | Std. Deviation | CLIC | CPIC | WL | KH |
|---|---|---|---|---|---|---|
| CLIC | 5.910 | 1.097 | 1 | | | |
| CPIC | 4.198 | 1.300 | −0.310 ** | 1 | | |
| NTB | 4.740 | 1.195 | −0.017 | 0.105 ** | | |
| WL | 2.841 | 0.986 | −0.469 ** | 0.477 ** | 1 | |
| KH | 4.639 | 1.233 | −0.461 ** | 0.445 ** | 0.450 ** | 1 |

Note: ** represents *p* < 0.05.

The relevant results show that there is a significant positive correlation between the following variables, and the difference is statistically significant: need to belong and competitive intrateam climate ($p \leq 0.01$, Pearson correlation coefficient = 0.105); workplace loneliness and competitive intrateam climate ($p \leq 0.01$, Pearson correlation coefficient = 0.477); knowledge hoarding and competitive intrateam climate ($p \leq 0.01$, Pearson correlation coefficient = 0.455); workplace loneliness and need to belong ($p \leq 0.01$, Pearson correlation coefficient = 0.198); and knowledge hoarding and workplace loneliness ($p \leq 0.01$, Pearson correlation coefficient = 0.450).

There is a significant negative correlation between the following variables, and the difference is statistically significant: competitive intrateam climate and collaborative intrateam climate ($p \leq 0.01$, Pearson correlation coefficient = −0.310); workplace loneliness and collaborative intrateam climate ($p \leq 0.01$, Pearson correlation coefficient = −0.469); and knowledge hoarding and collaborative intrateam climate ($p \leq 0.01$, Pearson correlation coefficient = −0.461).

There is no significant correlation between the following variables: need to belong and collaborative intrateam climate ($p > 0.05$); and need to belong and knowledge hoarding ($p > 0.05$).

## 4. Empirical Results

Based on the research hypothesis and the related analysis results presented above, the structural equation model of the latent variable was further established by Mplus (Model): The mediating effect of workplace loneliness on collaborative intrateam climate, competitive intrateam climate, and knowledge hoarding as well as the moderating effect of the need to belong on collaborative intrateam climate, competitive intrateam climate, and knowledge hoarding are discussed. The analysis results are as follows.

*4.1. The Mediating Effect of Workplace Loneliness on the Relationship between Collaborative Intrateam Climate, Competitive Intrateam Climate, and Knowledge Hoarding*

To investigate the mediating effect of workplace loneliness on collaborative intrateam climate, competitive intrateam climate, and knowledge hoarding, a latent variable structural equation model was established by Mplus, including collaborative intrateam climate and competitive intrateam climate as independent variables, workplace loneliness as a mediator variable, and knowledge hoarding as an outcome variable. Furthermore, the Bootstrap method was used to estimate the mediating effect size with 5000 iterations.

In this study, Mplus was used to establish a first-order six-factor confirmatory factor analysis model, and the fitting index of the structural equation model was: $\chi^2$ = 368.181, DF = 112, $\chi^2$/DF = 3.287 < 5, RMSEA = 0.060 < 0.08, SRMR = 0.041 < 0.08, CFI = 0.952 > 0.9, TLI = 0.942 > 0.9.

The fitting indexes of the structural equation model all reached the fitting standard. It was considered that the structural equation model could gain data support and the structure of the structural equation model was good. The analysis results are showed in Figure 2.

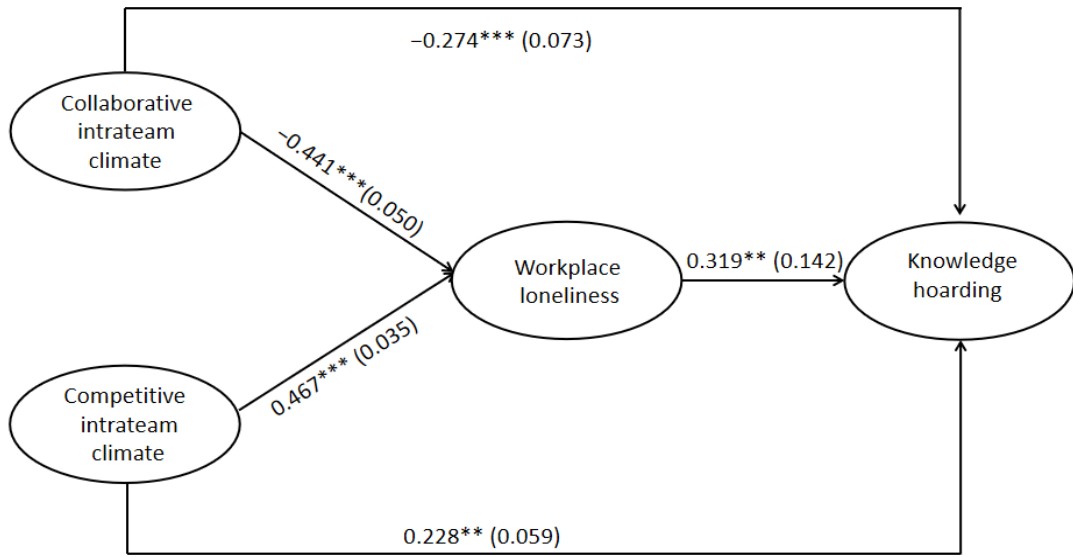

**Figure 2.** Structure variance model. Note: ** represents $p < 0.05$, *** represents $p < 0.01$.

The results of the structural equation analysis are shown in the table below (Table 5). First, the standardization coefficient of the measurement model is greater than 0.5, and the measurement model is well structured.

**Table 5.** Analysis results of structural equation model ($n = 639$).

| | Hypothesis Path | | | Estimate | Std. | S.E. | T-Value | *p*-Value | Result |
|---|---|---|---|---|---|---|---|---|---|
| H1a | CLIC | → | WL | −0.360 | −0.441 | 0.050 | −7.143 | 0.000 | Support |
| H1b | CPIC | → | WL | 0.300 | 0.467 | 0.035 | 8.458 | 0.000 | Support |
| H2a | CLIC | → | KH | −0.261 | −0.274 | 0.073 | −3.597 | 0.000 | Support |
| H2b | CPIC | → | KH | 0.171 | 0.228 | 0.059 | 2.904 | 0.004 | Support |
| H3 | WL | → | KH | 0.374 | 0.319 | 0.142 | 2.638 | 0.008 | Support |

Secondly, for the structural model, collaborative intrateam climate has a significant negative impact on workplace loneliness ($p < 0.001$, standardized path coefficient = −0.441); thus, hypothesis H1a is supported. Competitive intrateam climate has a significant positive impact on workplace loneliness ($p < 0.001$, standardized path coefficient = 0.467); thus, hypothesis H1b is supported. Collaborative intrateam climate has a significant negative effect on knowledge hoarding ($p < 0.001$, standardized path coefficient = −0.274); thus, hypothesis H2a is supported. Competitive intrateam climate has a positive effect on knowledge hoarding ($p = 0.004 < 0.05$, standardized path coefficient = 0.228); thus, hypothesis H2b is supported. Workplace loneliness has a positive effect on knowledge hoarding ($p = 0.008 < 0.05$, standardized path coefficient = 0.319); thus, hypothesis H3 is supported.

Further, the mediating effect between workplace loneliness and collaborative intrateam climate, competitive intrateam climate, and knowledge hoarding was discussed. The Bootstrap method was adopted with 5000 iterations to estimate the mediating effect size. The results are shown in Table 6.

For the mediating effect between workplace loneliness and collaborative intrateam climate and knowledge hoarding, the total effect, direct effect, mediating effect, and confidence interval do not contain 0, indicating that the total effect, direct effect, and mediating effect are significant. It is estimated that the standardized effect size of the total effect, direct effect, and mediation effect is −0.414, −0.274, and −0.140, respectively, and the percentage of total mediation effect is 33.8%. The analysis results show that the confidence interval of the mediating effect between workplace loneliness and collaborative intrateam climate and knowledge hoarding is [−0.293, −0.042], excluding 0. Workplace loneliness

has a significant mediating effect between collaborative intrateam climate and knowledge hoarding, and the mediating effect is −0.140; thus, hypothesis H4a is supported.

**Table 6.** Mediating effect estimation.

| Road | | Estimate | SE | 95%LCI | 95%UCI | Ratio (%) |
|---|---|---|---|---|---|---|
| CLIC→WL→KH | Total Effect | −0.414 | 0.050 | −0.513 | −0.321 | — |
| | Direct Effect | −0.274 | 0.074 | −0.409 | −0.115 | 66.2 |
| | Indirect Effect | −0.140 | 0.062 | −0.293 | −0.042 | 33.8 |
| CPIC→WL→KH | Total Effect | 0.377 | 0.050 | 0.278 | 0.476 | — |
| | Direct Effect | 0.228 | 0.077 | 0.083 | 0.381 | 60.5 |
| | Indirect Effect | 0.149 | 0.060 | 0.044 | 0.284 | 39.5 |

In terms of the mediating effect between workplace loneliness, competitive intrateam climate, and knowledge hoarding, the total effect, direct effect, mediating effect, and confidence interval do not contain 0, indicating that the total effect, direct effect, and mediating effect are significant. The estimated standardized effect size of the total effect, direct effect, and mediation effect were 0.377, 0.288, and 0.149, respectively, and the percentage of total mediation effect was 39.5%. The analysis results showed that the confidence interval of the mediating effect between workplace loneliness, competitive intrateam climate, and knowledge hoarding is [0.044, 0.284], excluding 0. Workplace loneliness has a significant mediating effect between the competitive intrateam climate and knowledge hoarding, and the mediating effect is 0.149; thus, hypothesis H4b is supported.

*4.2. The Moderating Effect of Need to Belong on Relationships between Collaborative/Competitive Intrateam Climates to Workplace Loneliness*

Based on the research results presented above, the moderating effect of the need to belong on workplace loneliness was further studied in terms of collaborative intrateam climate, competitive intrateam climate, and knowledge hoarding. The latent variable moderated mediation model was established by Mplus, including the independent variables of collaborative intrateam climate, competitive intrateam climate, and knowledge hoarding, including workplace loneliness as the mediating variable, need to belong as the moderating variable, and knowledge hoarding as the outcome variable. The model and analysis results are shown in Figure 3 and Table 7.

**Table 7.** The moderating effect of the need to belong on the mediation model.

| Hypothesis | Path | | | Estimate | Std. | S.E. | T-Value | *p*-Value | Result |
|---|---|---|---|---|---|---|---|---|---|
| H1a | CLIC | → | WL | −0.293 | −0.392 | 0.038 | −7.612 | 0.000 | Support |
| H1b | CPIC | → | WL | 0.237 | 0.402 | 0.032 | 7.402 | 0.000 | Support |
| H2a | CLIC | → | KH | −0.269 | −0.287 | 0.052 | −5.190 | 0.000 | Support |
| H2b | CPIC | → | KH | 0.180 | 0.244 | 0.042 | 4.261 | 0.000 | Support |
| H3 | WL | → | KH | 0.357 | 0.285 | 0.092 | 3.898 | 0.000 | Support |
| | NTB | → | WL | 0.215 | 0.264 | 0.039 | 5.454 | 0.000 | Support |
| H5a | CLIC × NTB | → | WL | 0.100 | 0.134 | 0.033 | 2.992 | 0.003 | Support |
| H5b | CPIC × NTB | → | WL | −0.100 | −0.169 | 0.031 | −3.231 | 0.001 | Support |

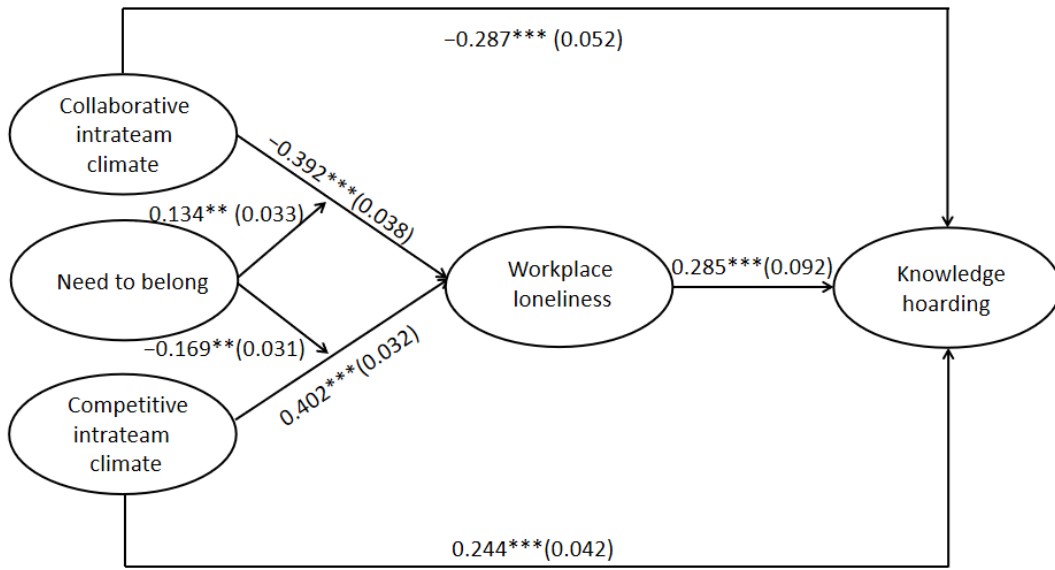

**Figure 3.** Moderated mediation model (*n* = 639). Note: ** represents *p* < 0.05, *** represents *p* < 0.01.

The results of the moderated mediation model analysis show that the need to belong has a significant positive moderating effect on collaborative intrateam climate and workplace loneliness (*p* = 0.003, standardization coefficient = 0.134); thus, hypothesis H5a is supported. The need to belong has a significant negative moderating effect on competitive intrateam climate and workplace loneliness (*p* = 0.001, standardization coefficient = −0.169); thus, hypothesis H5b is supported.

Take the simple slope tests to further test the moderated effect. The analysis results are shown in the following tables (Tables 8 and 9):

**Table 8.** Simple slope test.

|  | Estimate | S.E. | 95%LCI | 95%UCI |
|---|---|---|---|---|
| Low (−SD) | −0.619 | 0.051 | −0.718 | −0.520 |
| Median (0) | −0.392 | 0.040 | −0.471 | −0.313 |
| High (+SD) | −0.165 | 0.052 | −0.267 | −0.063 |
| High–Low | 0.453 | 0.064 | 0.329 | 0.578 |

**Table 9.** Simple slope test.

|  | Estimate | S.E. | 95%LCI | 95%UCI |
|---|---|---|---|---|
| Low (−SD) | 0.535 | 0.047 | 0.444 | 0.627 |
| Median (0) | 0.317 | 0.035 | 0.249 | 0.386 |
| High (+SD) | 0.099 | 0.045 | 0.012 | 0.187 |
| High–Low | −0.436 | 0.059 | −0.551 | −0.321 |

As shown in Table 8, when the moderating variable is lower than one standard deviation, the collaborative intrateam climate has a significant negative impact on workplace loneliness, with a 95% confidence interval [−0.718, −0.520] and effect size of −0.619. When the moderating variable is one standard deviation higher, the collaborative intrateam climate has a significant negative impact on workplace loneliness, with a 95% confidence interval [−0.267, −0.063] and effect size −0.165. Under the moderation of high and low moderating variables, there is a significant difference between collaborative intrateam climate and workplace loneliness, with a 95% confidence interval [0.329, 0.578] and a difference value of 0.453. The results show that with the increase of the need to belong, the negative impact of collaborative intrateam climate on workplace loneliness gradually weakened.

The simple slope test was further conducted, and the analysis results show that when the moderating variable is lower than one standard deviation, the competitive intrateam climate has a significant positive effect on workplace loneliness, with a 95% confidence interval [0.444, 0.627] and effect size of 0.535. When the moderating variable is one standard deviation higher, competitive intrateam climate has a significant positive impact on workplace loneliness, with a 95% confidence interval [0.012, 0.187] and an effect size of 0.099. Under the moderation of high and low moderating variables, there is a significant difference between competitive intrateam climate and workplace loneliness, with a 95% confidence interval [−0.551, −0.321] and a difference value of −0.436. The results show that with the increase of the need to belong, the positive impact of competitive intrateam climate on workplace loneliness gradually weakened.

In order to more intuitively reflect the moderating effect of the need to belong on collaborative intrateam climate within the team and workplace loneliness, the regression coefficient is used as the moderating effect, as shown in the figure below:

Figure 4 shows the slope comparison of the impact of teamwork atmosphere on workplace loneliness under the adjustment of the need to belong. The results show that the slope of collaborative intrateam climate on workplace loneliness under the regulation of low need to belong is greater than that under the regulation of a high need to belong. That is, with the increased need to belong, the negative effect of the cooperative atmosphere on workplace loneliness gradually decreases.

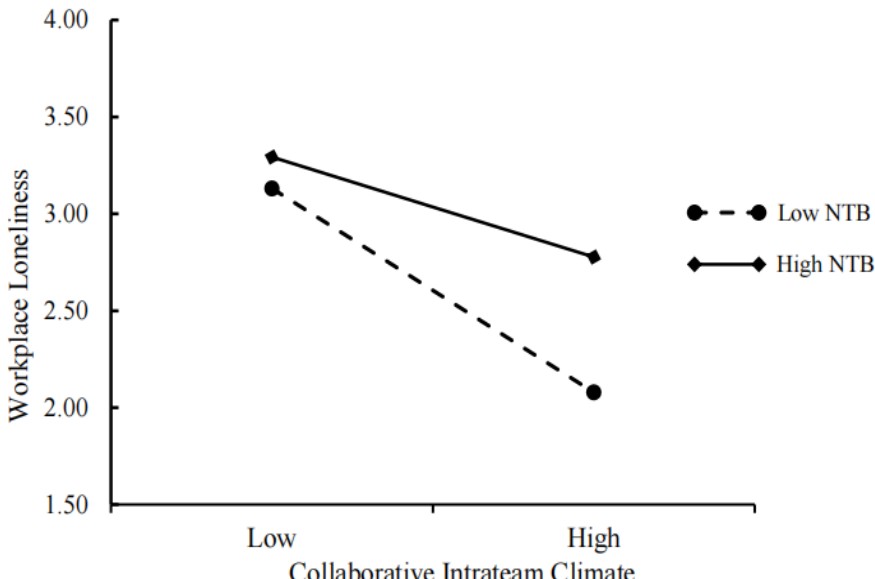

**Figure 4.** The moderating role of NTB on the relationship between CLIC and WL.

As shown in Figure 5, under the regulation of a low sense of needing to belong, the slope of competitive intrateam climate on workplace loneliness is greater than that under the regulation of the high sense of needing to belong. That is, with the increasing of need to belong, the positive impact of competitive intrateam climate on workplace loneliness gradually weakens.

### 4.3. The Moderating Effect of Need to Belong on the Mediation Model

The coefficient product method proposed by Hayes and the difference test of mediating effect size under high and low regulation proposed by Edwards were further used to test the moderated mediating effect [77]. In other words, the significance of the product of path coefficients between interaction terms and mediating variables was tested to determine whether the moderated mediating effect is significant and whether it remains significant under the high-low moderating effect. The 95% confidence interval of the difference does

not include 0, so the moderated mediating effect is significant. The analysis results are shown in the following table (Table 10):

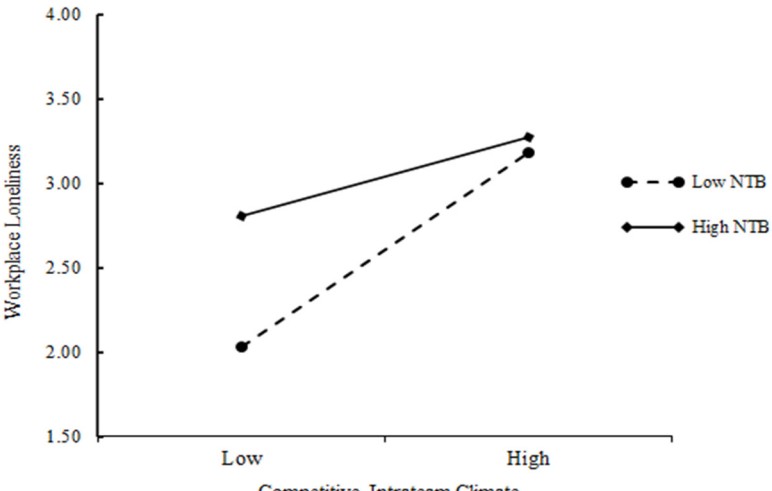

**Figure 5.** The moderating role of NTB on the relationship between CPIC and WL.

**Table 10.** Moderated mediating effect estimation.

| Road | | Estimate | SE | 95%LCI | 95%UCI |
|---|---|---|---|---|---|
| CLIC→WL→KH | Low (−SD) | −0.140 | 0.039 | −0.216 | −0.064 |
| | Median (0) | −0.104 | 0.030 | −0.163 | −0.046 |
| | High (+SD) | −0.069 | 0.026 | −0.120 | −0.018 |
| | High–Low | 0.072 | 0.028 | 0.016 | 0.127 |
| | Moderated Mediation | 0.036 | 0.014 | 0.008 | 0.063 |
| CPIC→WL→KH | Low (−SD) | 0.120 | 0.035 | 0.051 | 0.190 |
| | Median (0) | 0.085 | 0.025 | 0.036 | 0.133 |
| | High (+SD) | 0.049 | 0.020 | 0.010 | 0.087 |
| | High–Low | −0.071 | 0.029 | −0.129 | −0.014 |
| | Moderated Mediation | −0.036 | 0.015 | −0.064 | −0.007 |

The results show the following. As for the moderating effect of moderating variable attribution need on the mediating effect of workplace loneliness on collaborative intrateam climate and knowledge hoarding within the team. The difference between the high-low moderating variables was 0.072, the 95% confidence interval was [0.016, 0.127], and the confidence interval did not include 0, indicating that there was a significant moderating mediating effect in the high-low moderating variables. The coefficient product method was further used to test the moderated mediating effect size, and the analysis results show the following. The moderating mediating effect size was 0.036, the standard error was 0.014, and the confidence interval was [0.008, 0.063], indicating that need to belong positively moderated the mediating effect between workplace loneliness and collaborative intrateam climate and knowledge hoarding; hence, hypothesis H6a was supported.

As for the moderating role of the need to belong on the mediating effect of workplace loneliness between competitive intrateam climate and knowledge hoarding, the difference between the high-low moderating variables was −0.071, and the 95% confidence interval was [−0.129, −0.014]. The confidence interval did not contain 0, indicating a significant difference in the mediating effect size in the high-low moderating variables. The coefficient product method was further used to test the moderated mediating effect size, and the analysis results show the following. The moderating mediating effect size was −0.036, the standard error was 0.015, and the confidence interval was [−0.064, −0.007], indicating

that need to belong had a negative moderating effect on the mediating effect of workplace loneliness on competitive intrateam climate and knowledge hoarding; hence, hypothesis H6b was supported.

*4.4. Summary of Hypotheses*

In Table 11, we summarize the results of the whole analyses from hypotheses 1a to 6b. We find that all of the hypotheses are supported.

**Table 11.** Summarize of Hypotheses 1a–6b.

| | **Hypotheses** | **Results** |
|---|---|---|
| H1a | A collaborative intrateam climate is negatively associated with workplace loneliness. | Support |
| H1b | A competitive intrateam climate is positively associated with workplace loneliness. | Support |
| H2a | A collaborative intrateam climate is negatively associated with knowledge hoarding. | Support |
| H2b | A competitive intrateam climate is positively associated with knowledge hoarding. | Support |
| H3 | Workplace loneliness is positively associated with knowledge hoarding. | Support |
| H4a | Workplace loneliness mediates the relationship between a collaborative intrateam climate and knowledge hoarding. | Support |
| H4b | Workplace loneliness mediates the relationship between a competitive intrateam climate and knowledge hoarding. | Support |
| H5a | Need to belong weakens the negative relationship between collaborative intrateam climate and workplace loneliness. | Support |
| H5b | Need to belong weakens the positive relationship between competitive intrateam climate and workplace loneliness. | Support |
| H6a | Need to belong positively moderates the mediating effect of workplace loneliness on the relationship between collaborative intrateam climate and knowledge hoarding. | Support |
| H6b | Need to belong negatively moderates the mediating effect of workplace loneliness on the relationship between competitive intrateam climate and knowledge hoarding. | Support |

## 5. Discussion

This paper reviews the conceptual frameworks of SET, KST, the need to belong hypothesis, and relative deprivation theory to explain intrateam climate, workplace loneliness, and knowledge hoarding. Negative emotions in the workplace can easily lead to a lack of good relationships and negative behaviors. Loneliness in the workplace is associated with negative emotions [78]. With the change of working methods and evolution of employee division of labor, more work is completed by individuals independently, resulting in the reduction of cooperation opportunities, which weakens the communication ability between employees [7]. Furthermore, internet telecommuting has been shown to increase the feeling of loneliness in the workplace.

This paper theoretically considers the factors of team situation, introduces the concepts of collaborative and competitive intrateam climate, and deduces the hypothesis that employees' workplace loneliness produces knowledge hoarding under the influence of team climate. Since loneliness is an emotion associated with intimacy, we introduce the need to belong, whose core motivation is the desire to pay attention, accept, and like. The need to belong partly drives our preference to be with others [38]; thus, workplace loneliness can be effectively adjusted.

*5.1. Theoretical and Empirical Implications*

Firstly, as revealed by the review of existing literature, there is little research on workplace loneliness in the context of Chinese culture, especially during environmental

disruptions such as COVID-19, which means that not enough attention has been paid to this issue. More consideration should be given to the mental health problems of employees in the workplace [1]. Through an exploration of knowledge hoarding, a common but rarely studied outcome variable, this study addresses the negative impact of workplace loneliness and looks for ways to alleviate it.

Secondly, previous studies on workplace loneliness have mostly focused on the relationship between the two variables, without an in-depth discussion of its mediating mechanism [1]. This study examines the mediating effect of workplace loneliness and introduces the need to belong as the moderating variable; reveals the formation mechanism of workplace loneliness in the collaborative and competitive intrateam climate; and deepens the research on the effective regulation of workplace loneliness.

Third, we conducted a study on the need to belong in the organizational context, especially during the COVID-19 crisis. Based on need to belong theory, we integrate the need to belong, team climate, and workplace loneliness into a single model. We verified that the need to belong weakens the relationship between collaborative/competitive intrateam climate and workplace loneliness. Simultaneously, the negative/positive mediating effects of workplace loneliness on the relationship between collaborative/competitive intrateam climate and knowledge hoarding weakens when the need to belong increases. This conclusion contributes to the theory of belonging, especially under COVID-19.

### 5.2. Practical Implications

The hypotheses of this study are supported by the data results and provide several useful insights for employee management.

Firstly, employees should be aware of the harm caused by workplace loneliness [1], especially under environmental disruptions such as COVID-19. A survey of 2700 employees found that since the pandemic, 75% said they felt more isolated socially, 67% felt more stressed and 57% said they had higher levels of anxiety [79,80]. This study confirms that controlling loneliness is important for an organization [81]. In addition, the importance of exchange relationships between employees and colleagues in forming a link between loneliness and knowledge hoarding was reexamined. Activities and training programs to improve social relations need to be provided at the organizational level. In order to enable employees to establish organic social relationships with their colleagues, the organization should strive to remove obstacles for individuals to establish organizational ties [82], providing employees with a place for long-term emotional communication and benefiting the organization in the long term.

Secondly, employees' poor mental health may affect the overall performance of the organization, especially under environmental disruptions such as COVID-19. The pandemic also highlights the need for organizations and leaders to listen to employees' voices in order to better understand their physical and mental health risks [5] and the factors that affect or potentially affect employees' mental health [83]. In particular, the organization should support employees' emotional intelligence, mental health, and positive psychological capital to prevent employees from being lonely [83]. In this case, the organization should carry out practice and training and strengthen cooperation among employees by establishing an organizational atmosphere of mutual help, which will reduce employees' loneliness.

Thirdly, our research shows that the need to belong can effectively moderate the negative impact of team climate on workplace loneliness in the era of COVID-19. A sense of belonging can be used as a form of treatment to help those who are often or long-term excluded [84]. The role of belonging in psychotherapy is also reflected in the effectiveness of group therapy. As Lewin clearly pointed out, "It is easier to change individuals formed into a group than change them separately" [85]. Organizations can implement sense of belonging support programs, including training and interaction opportunities [25]. Organizations should also create an environment conducive to individuals' pursuit of a sense of belonging [64].

Fourth, knowledge hoarding shows moral failure, and knowledge hoarders need to be taught to care more about their colleagues [86]. It is usually considered a disease in the organizational information ecosystem, as it has a negative impact on the organization's knowledge management practice and information ability. Although the decision to hoard knowledge is usually rational and reasonable from an individual perspective, it is destructive from an organizational perspective [47,87]. Successful knowledge management projects rely on sharing rather than hoarding [88]. Therefore, management should make employees understand that sharing knowledge is more valuable than hoarding it [47], especially during environmental disruptions such as COVID-19.

Finally, although our empirical research results indicate that a competitive intrateam climate leads to workplace loneliness, we believe that cooperation is not the only positive type of intrateam climate. In some cases, competition may be an effective management technology [89]. Competition may increase external motivation, leading to a higher level of creativity. When the tasks or projects involved are not intellectually challenging and cannot create their own interests, a competitive climate may be effective [32]. From this perspective, we suggest using the competitive climate carefully and call for an in-depth analysis of the dynamic mechanism of the competitive atmosphere in future research.

### 5.3. Limitations and Future Research

Several potential limitations of this study are worth mentioning.

First, we can only get the subjective results because all measurements involved self-reports of participants' perception of climate, workplace loneliness, knowledge hoarding, and need to belong. We still need to do more work to study the behavioral and even physiological differences of people with different attribution motives. A theoretical article once proposed that many gender differences are based on the fact that women are eager to connect while men are not. Baumeister expressed that it is not that men do not care about social connections. The evidence shows that men prefer larger social groups while women give priority to one-on-one intimacy. He once showed in the interview that Gardner and Gabriel's subsequent work supported this conclusion [67].

Secondly, over time, we cannot observe the hoarding and detention behavior of the same subject in different periods, which is a disadvantage that cannot be solved with a single cross-sectional data set. Longitudinal data may be needed to capture the true nature of hoarding behavior [18]. In addition, all variables in this study are from employee self-assessment, which may lead to the problem of method deviation. Therefore, we encourage future research to use different sources, including colleagues and supervisors, to assess the organizational bias of focus employees [64].

Third, the results of this study may give only a partial view because it is a survey conducted only in China. Although the ideal amount of data was obtained, the subjects' workplaces were scattered, and the industries were diverse. Chinese culture has a high degree of collectivism, so Chinese people are more vulnerable to the influence of a sense of belonging than people in Western countries. It is advisable to compare Chinese and Western samples in cross-cultural research [64]. It would be interesting for future research to test whether our results also apply to Western individualist societies.

Fourth, this study is limited by using only quantitative methods and only obtaining data through questionnaires. Creative methods may be needed (such as anonymous online interviews and creating a private space outside of work to discuss experiences) [27]. In the future, researchers can use better research methods and designs to draw reliable conclusions. We recommend that future researchers use both quantitative and qualitative methods to collect and interpret data. Qualitative methods can supplement the results of the survey and help to understand the factors related to loneliness. For example, researchers may use video observations, focus group interviews, and laboratory experiments to gain insight into human emotions. Video observation can record not only audio-visual data but also capture facial expressions, gestures, postures, and body movements in the process of interaction,

which helps researchers to go deep into a new level of emotional experience from the perspective of psychology and neurobiology [3].

Finally, this study makes assumptions about the characteristics of the times. Kotera et al. showed that with the widespread of COVID-19 in 2019, the mental health problems and loneliness of Japanese medical staff were higher than those of the general population [90,91]. More research can be carried out in the future, such as the contribution of physical distance and social isolation to loneliness in an organization, which may lead to a richer understanding of the processes by which meaningful relationships are inhibited (such as working from home, telecommuting, and "casual" economic work). Under these lasting conditions, loneliness may become more long-term for individuals, forming a permanent downward cycle. However, more research is needed to isolate the conditions that cause this social isolation (self-selection or organization-imposed) and related buffer variables (such as virtual connection or meaningful job compensation) [27].

## 6. Conclusions

Using our sample of 639 respondents in the era of COVID-19, this paper establishes a latent variable structural equation model through Mplus to explore the occurrence path and antecedents of workplace loneliness and knowledge hoarding. Specifically, collaborative and competitive intrateam climates affect employees' workplace loneliness and knowledge hoarding from different aspects. This paper examines the mediating role of workplace loneliness and the moderating role of the need to belong. Team climate causes employees' knowledge-hoarding behavior through workplace loneliness. At the same time, we found that the need to belong can well moderate workplace loneliness, make the collaborative intrateam climate more conducive to sharing, and alleviate the penetration of the competitive intrateam climate into workplace loneliness. Finally, we verified a moderated mediation model.

These findings remind organization leaders that when optimizing the allocation of relationship resources, they should take into account the future vision of the organization and current mental health and emotional state of employees. At the same time, the relationship between employees in the organization should be improved by appropriately adjusting the climate of group competition and cooperation.

In the era of COVID-19, people's life and work are disrupted. The rapid spread and high death rate make us feel unstable and vulnerable. People tend to become anxious and lonely because of the change in lifestyle and the decline in income, which is the reason why we investigated a series of questions in this study. The remote office environment makes it more convenient and easier for us to collect first-hand questionnaires through the network, but as analyzed above, we still have limitations in the selection of samples. We tried our best to develop a solid theoretical foundation, and the data results well verify the original assumption. Future research can consider more possible moderators, especially in relation to psychological variables in terms of organizational behavior. It will be very interesting to further explore the positive impact of workplace loneliness in this vein.

**Author Contributions:** Formal analysis, Y.M.; Methodology, Y.M.; Supervision, S.D. and J.Y.L.; Writing—original draft, Y.M.; Writing—review & editing, J.Y.L. All authors have read and agreed to the published version of the manuscript.

**Funding:** This research received no external funding.

**Institutional Review Board Statement:** Ethical review and approval were waived for this study because we secured the confidentiality of all respondents, and the questionnaire items were made based on very common items by following the established studies.

**Informed Consent Statement:** Informed consent was obtained from all subjects involved in the study.

**Conflicts of Interest:** The authors declare no conflict of interest.

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
