# Peer review of "Workplace Loneliness and the Need to Belong in the Era of COVID-19"

_sustainability, doi:10.3390/su14084788_

Round 1
Reviewer 1 Report
The article deals with an interesting and important issue; however, I had some concerns about the manuscript which, in its current form, limit the contribution it can make. I detail these below.
Introduction:
Hypotheses: The unconditional causal wording, given the non-experimental method is, for me, a major concern. The authors should be careful not to claim causality. For example: Instead of H1b A competitive intrateam climate positively impacts workplace loneliness, should be: A competitive intrateam climate is positively associated with workplace loneliness.
The hypotheses 5a and 5b should be reformulated as phrases like „positively moderates” or negatively moderates” are not accurate enough. Instead, the authors should describe how the moderator will affect the relationship between X and Y.
In addition, I suggest the authors provide a more detailed rationale for their hypotheses regarding the moderating effects.
It is also not clear how the mediation effects would be moderated by need to belong.
Empirical results:
The figures 4 and 5 are incomplete - the names of variables are overlooked on the graphs; both levels of moderators should be depicted on the graphs’ legends.
All moderating analyses should contain the calculations of conditional effects (simple slope tests). The authors provided them only for the moderated mediating effect (see Table 8).
Discussion:
The discussion of the results is too superficial, especially when it comes to the interpretation of the both moderating and moderated mediation effects.
In addition, the effect depicted in Figure 5 is described incorrectly (P. 15, l. 512-515 – “As shown in Figure 5, the results of the slope comparison of the effect of collaborative intrateam climate on workplace loneliness under the regulation of the need to belong show that the slope of the effect of a competitive atmosphere on workplace loneliness under the regulation of need to belong is higher than that under the regulation of need to belong”).
Author Response
Please see our attached response letter. Thank you very much.

Reviewer 2 Report
Congratulate the author(s) for the work done. It seems to me a relevant, current and necessary work. I believe, without a doubt, that it should be published and is relevant for the international scientific community. However, for its better visibility and readability, I make some minimal constructive recommendations, with the best academic spirit:
The title is interesting and striking, but I think too general or not fully representative of what is in the article. I believe that broadening it, to the extent permitted by the journal's rules, will help its better indexing, readership and positioning. If possible, I would slightly increase its length.
The abstract is comprehensive, contains all the important sections and is well developed. However, when using acronyms, the first word should be capitalized (check in the abstract and in the text, the first time acronyms are used).
The keywords are very well chosen. However, since they are formed by conjunctions of two words, it can harm the searches and its positioning in databases. If it were possible to add more keywords, I would recommend adding some more, which are simple and more general, so that they can be used more easily in searches.
The papers cited as theoretical framework are solid and prestigious but some are very old and there are few recent and international citations for a journal of this prestige. I recommend adding 5-6 super up-to-date references, from top international journals, especially when a paper is being offered on this important topic after the COVID-19 crisis. Those new references should be only from 2021 and 2022.
The Results and Discussion sections are brilliant, as the exposition and argumentation are very well exposed and spun.
It is necessary to develop a section on limitations and prospects, in a separate section, at the end: why there is a problem (new, current, original) to investigate, why this methodology has been chosen and what limitations the chosen sample has, why there are conclusions that generate an original advance in knowledge, how this study can be replicated by the academic community.
A harmonious distribution of paragraphs should be sought, that they all have a similar length, 6-7 lines, with neither very long paragraphs, nor very short paragraphs. This will make the text more readable and understandable even if it is already well written, since there are very long paragraphs at the end, unbalanced compared to those of the introduction, which are balanced.
Author Response
Please see the attached response letter. Thank you very much.
